# Preclinical Testing of Boron-Doped Diamond Electrodes for Root Canal Disinfection—A Series of Preliminary Studies

**DOI:** 10.3390/microorganisms10040782

**Published:** 2022-04-07

**Authors:** Maximilian Koch, Victor Palarie, Lisa Koch, Andreas Burkovski, Manuel Zulla, Stefan Rosiwal, Matthias Karl

**Affiliations:** 1Division of Microbiology, Department of Biology, Friedrich-Alexander-Universität Erlangen-Nürnberg, 91058 Erlangen, Germany; max.koch@fau.de (M.K.); lisa.koch@fau.de (L.K.); andreas.burkovski@fau.de (A.B.); 2Laboratory of Tissue Engineering and Cellular Cultures, State University of Medicine and Pharmacy “N. Testemitanu”, 2004 Chisinau, Moldova; vpalarie@gmail.com; 3Division of Materials Science and Engineering for Metals, Department of Material Sciences, Friedrich-Alexander-Universität Erlangen-Nürnberg, 91058 Erlangen, Germany; manuel.zulla@fau.de (M.Z.); stefan.rosiwal@fau.de (S.R.); 4Division of Prosthodontics, Saarland University, 66421 Homburg, Germany

**Keywords:** antimicrobial treatment, canine tooth model, electrochemical disinfection, endodontics, reactive oxygen species

## Abstract

While numerous approaches have meanwhile been described, sufficient disinfection of root canals is still challenging, mostly due to limited access and the porous structure of dentin. Instead of using different rinsing solutions and activated irrigation, the electrolysis of saline using boron-doped diamond (BDD) electrodes thereby producing reactive oxygen species may be an alternative approach. In a first step, experiments using extracted human teeth incubated with multispecies bacterial biofilm were conducted. The charge quantities required for electrochemical disinfection of root canals were determined, which were subsequently applied in an animal trial using an intraoral canine model. It could be shown that also under realistic clinical conditions, predictable disinfection of root canals could be achieved using BDD electrodes. The parameters required are in the range of 5.5 to 7.0 V and 9 to 38 mA, applied for 2.5 to 6.0 min with approximately 5 to 8 mL of saline. The direct generation of disinfective agents inside the root canal seems to be advantageous especially in situations with compromised access and limited canal sizes. The biologic effect with respect to the host reaction on BDD-mediated disinfection is yet to be examined.

## 1. Introduction

Removing vital and necrotic remnants of pulp tissues, microorganisms and microbial toxins constitutes the most important step in root canal treatment [1,2,3]. However, due to the variability of root canal morphology [4,5], the presence of irregularities [3,6,7] and the fact that the accessible canal system is much smaller than the total open volume present inside a tooth [8], it can be considered as being impossible to completely shape and clean the root canal [8]. This is also supported by a previous study showing that bacterial biofilms penetrate deep within dentin tubules [9]. As a consequence, microbes can remain at least in infected dentin, regrow and cause inflammatory processes [10,11]. As such, a strong association between persistent intraradicular infection and post-treatment apical periodontitis has been described [12,13].

With the aim of improving removal of microbes from root canals, irrigation protocols have been optimized [14,15] by activating irrigation solutions in order to also reach dentin canals and lateral canals. However, it has recently been shown for ultrasonic activation systems to cause smear layers in the apical parts of the canal system [16], counteracting the desire of removing bacterial biofilms, which are more resistant than their planktonic counterparts [17]. In addition, a wide variety of advanced treatment modalities have been introduced including continuous apical negative ultrasonic irrigation [18,19], high-frequency electrical pulses [20], electric current and silver electrodes [21] as well as Er:YAG laser irradiation [22]. Most of these techniques result in a decrease in bacterial numbers but cannot predictably prevent bacterial regrowth [23]. In this context, electrophoresis of copper/calcium hydroxide-based endodontic paste also constitutes an approach, which uses electric current for disinfecting root canals. Two recent publications have pointed out that positive results may be obtained with this technique [24,25]. 

Given the impossibility of sterilizing the canal system of a tooth and the porous structure of surrounding dentin, it was hoped that root canal obturation would inactivate or prevent bacteria from repopulating the canal space [26]. Despite different treatment modalities and materials being available, current concepts can neither prevent leakage [27], nor bacterial growth on obturation materials [28,29]. Instead, coronal exposure of root canal obturation has been claimed to require retreatment due to microbial contamination [30] and even gutta-percha points with antimicrobial properties cannot be seen as a solution [31].

Electrochemical disinfection using boron-doped diamond (BDD) electrodes has previously been described as being efficient in inactivating bacteria and yeasts both in endodontic and periimplantitis treatment [32,33,34,35]. So far, this technology has been tested under controlled laboratory conditions using single species biofilms instead of using more realistic multispecies biofilms [12]. Additionally, fully instrumented root canals with enlarged lumen able to accommodate prototype BDD electrodes have been considered. These canals were contaminated after enlargement subsequently only using BDD electrodes for disinfection. However, in clinical practice, shaping of root canals is performed after contamination, e.g., through a caries lesion, and helps to mechanically remove biofilm and infected dentin.

After constructing a clinically applicable disinfection apparatus, it was the goal of the experiments to define performance parameters for electrochemical disinfection in an in vitro setting. Based on these findings, electrochemical disinfection of root canals was tested in an animal experiment.

## 2. Materials and Methods

### 2.1. BDD Electrode Treatment Systems

Niobium wires of 200 µm thickness for prototype 1, were produced as described previously [32,35]. Niobium wires with a 50 µm diameter for prototype 2 (99.9% Nb-containing, Nb-502, Haines & Maassen Metallhandelsgesellschaft mbH, Bonn, Germany) were pre-treated with sandblasting (5 bar, Cemat NT4, Wassermann Dental-Maschinen GmbH, Hamburg, Germany) using silicon carbide particles (17–74 µm, SiC F320) for optimal properties with respect to surface roughness for diamond adhesion and stiffness. Sandblasted wires were cleaned in an ultrasonic bath (2 min, 45 kHz, Fa. Elma, Elmasonic X-tra) and diamond was seeded on the surface using nanodiamond dispersions (1:1000 or 1:10,000 in ethanol, Andante, Carbodeon Ltd. Oy, Vantaa, Finland). Diamond coating with boron doping was performed in a Hot-Filament Chemical Vapor Deposition machine (HFCVD). Tungsten filaments (220 mm, Ø 100 µm) were fit into a filament mount and pre-heated for carburization (18 h, 65 A/mount) in order to reach stable conditions during the deposition process following the pre-heating process in a methane–hydrogen–trimethyl borate gas atmosphere [32]. 

### 2.2. Determination of Disinfection Parameters

Eight extracted single-rooted human teeth were cut at the cemento-enamel junction using a diamond band saw (EXAKT 300, EXAKT Advanced Technologies GmbH, Norderstedt, Germany). The root canals were hand-instrumented using conventional Hedstroem files and intermediate irrigation with chlorhexidine solution (CHX, Chlorhexamed FORTE alkoholfrei 0.2%, GlaxoSmithKline Consumer Healthcare, Munich, Germany) until ISO #50 was reached. Subsequently, specimens were autoclaved, embedded in silicone (Aquarium-Dicht, Bindulin-Werk H.L. Schönleber GmbH, Fürth, Germany) and biofilm formation was induced by inoculation with *Enterococcus faecalis* (DSM 20478), submerging the roots in brain heart infusion (BHI, Oxoid, Wesel, Germany) and incubating them for 24 h at 37 °C. Teeth were rinsed with saline (0.9% NaCl). 

Before disinfection, sampling of bacteria using absorbent paper points was carried out [36]. As control, specimens were rinsed with saline for 0.5 and 7.5 min (2 mL/min). Electrochemical disinfection was performed using prototype 1 (Figure 1a) allowing for electrolyte and electrical power application inside the root canal (Figure 1b) using a combination of a cannula and external BDD wire (Figure 1c). 

The test specimens were treated for 0.5, 1.0, 1.5, 2.5, 5.0 and 7.5 min using 2 mL/min of saline irrigation with electric power of 20 mA and approximately 6–7 V applied to the BDD electrodes as derived from experiments carried out previously [32,33,34]. Paper point sampling was performed after treatment as described above. The paper points were transferred to BHI (1 mL, 5 min), and a decimal dilution series was plated on BHI agar plates, which were incubated overnight (37 °C) followed by determination of colony forming units (CFU).

### 2.3. Analysis of Multispecies Biofilm Formation

A total of nine extracted multi-rooted human teeth were cut at the cemento-enamel junction using a diamond band saw (EXAKT 300, EXAKT Advanced Technologies GmbH, Norderstedt, Germany). The root canals were hand-instrumented using conventional Hedstroem files and intermediate irrigation with chlorhexidine solution (CHX, Chlorhexamed FORTE alkoholfrei 0.2%, GlaxoSmithKline Consumer Healthcare, Munich, Germany) until ISO #50 was reached. Subsequently, specimens were autoclaved twice and stored in saline solution until further use.

Three teeth were split in half and all teeth or tooth halves were submerged in four milliliters of brain heart infusion (BHI, Oxoid, Wesel, Germany), respectively. The teeth were inoculated with multispecies biofilm obtained from a previous experiment (Ethik-Kommission der Friedrich-Alexander-Universität Erlangen-Nürnberg, Project 159_16B, Erlangen, Germany) and incubated for five or seven days at 37 °C and 175 rpm. Subsequently, the roots were longitudinally split and washed three times with phosphate-buffered saline (PBS). Potential biofilm was isolated from root canals using toothpicks and placed on glass slides for microscopy or tooth segments were glued onto glass slides before samples were stained using EbbaBiolight 680 (Ex/Em 530–565 nm/600–800 nm, Ebba Biotech AB, Solna, Sweden). After thirty minutes of incubation, the samples were gently washed with water, dried and analyzed with a confocal laser scanning microscope (CLSM, Axio Vert.A1, Carl Zeiss Microscopy Deutschland GmbH, Oberkochen, Germany) using the following equipment: Objectives: A-Plan 10×/0.25 Ph 1; LD A-Plan 40×/0.55 Ph 1; EC Plan-Neofluar 63×/1.25 Oil M27; Channel: mRFP, bright; Camera: AxioCamICm1; Software: Zen 2.3 (blue edition, Version 2.3.69.1018).

### 2.4. Disinfection of Root Canals Colonized by Multispecies Biofilm

Fifteen extracted human teeth were cut as described above. In the case of multi-rooted teeth, the roots were separated such way that only single roots with single root canals were present in conventional radiographs. All root canals were hand-instrumented until a glide path ISO #20 could be established followed by sterilization using an autoclave.

In a previous experiment [8,29], the aerobic bacterial flora of teeth requiring root canal treatment was determined and used for culturing a multispecies biofilm (Ethik-Kommission der Friedrich-Alexander-Universität Erlangen-Nürnberg, Project 159_16B, Erlangen, Germany). Bacterial suspensions in BHI derived from this culture were applied into the root canal systems followed by incubation at 37 °C for seven days.

The outer surfaces of the roots were cleaned using a mixture of 32.5% isopropanol, 18% ethanol, 0.1% glutaraldehyde and 49.4% distilled water [37] and the roots were positioned in sterile plasticine [38] in order to allow for root canal shaping using reciprocating instruments (R25 RECIPROC, VDW GmbH, Munich, Germany) until reaching the apices. The specimens were split into five treatment groups (n = 3 per group), applying the protocols detailed in Table 1 as final treatment steps.

Following treatment, bacterial sampling was carried out as described above (see Section 2.2). The roots were split longitudinally, and the segments were pressed on blood agar plates, which were incubated overnight at 37 °C, as previously described [32]. 

Selected colonies found on blood agar plates following incubation of paper points were subject to matrix-assisted laser desorption ionization time of flight (MALDI-ToF) mass spectrometry (Ripac-Labor, Potsdam, Germany) for identification of the species surviving the different forms of root canal disinfection. 

### 2.5. Canine Tooth Model

Following ethics commission approval (Comitetului de Etica a Cercetarii, State Medical and Pharmaceutical University “Nicolae Testemitanu”, Chisinau, Moldova), four dogs (breed Jack Russel Terrier) with a minimum age of 24 months were allocated for this study. All animals underwent two interventions, which consisted of trepanation of selected maxillary incisors (combined with bilateral extractions of mandibular premolars and molars; experiment not reported here) followed by root canal treatment of infected and non-infected teeth three weeks later (combined with bone augmentation procedures in the mandible; not reported here).

General anesthesia was induced and maintained using an intravenously administered combination of Xylacin (Xyla 20 mg/mL, 0.15 mL/kg, Interchemie werken ‘De Adelaar’, Waalre, The Netherlands) and Ketamin (Ketamin-hameln 50 mg/mL, 0.1 mL/kg, hameln pharma plus GmbH, Hameln, Germany). Heart rate, respiratory rate, O_2_ saturation and expiratory CO_2_ were monitored throughout all surgical procedures (Low Flow Capnograph V900040LF, SurgiVet Inc, Waukesha, WI, USA). Upon finishing the interventions, atipamezole hydrochloride was administered for recovery of the animals (Antimedin, ZOOFARMAGRO SRL, Chişinău, Moldova). Local anesthetic was applied (UDS Forte, Sanofi, Frankfurt am Main, Germany) prior to any intervention followed by intraoral disinfection using chlorhexidine (Chlorhexamed FORTE alkoholfrei 0.2%, GlaxoSmithKline Consumer Healthcare, Munich, Germany). The animals were kept as a group in a controlled facility and were fed with soft food and water ad libitum. Six weeks after the second intervention, the animals were sacrificed by injection of T-61 Euthanasia Solution (0.12 mL/kg; Merck Animal Health, Madison, NJ, USA) after inducing general anesthesia as described above and block sections of both jaws were harvested for histologic analysis (results not reported here).

Root canal treatment of maxillary incisors was carried out applying conventional chemo-mechanical protocols using Hedstroem files and CHX solution for rinsing. Intraoral radiographs (60 kV, 7 mA, 0.5 s; Heliodent, Dentsply Sirona, York, PA, USA; Ergonom X, Dentalfilm, Settimo Torinese, Italy) were used for determining working length. In addition, BDD electrodes (Figure 2) were applied following the shaping process using following parameters: 5.5 to 7.0 V and 9 to 38 mA applied for 2.5 to 6.0 min with approximately 5 to 8 mL of saline for irrigation.

Microbial sampling was performed after trepanation, after complete chemo-mechanical instrumentation and after BDD application either using endodontic files or paper points. These samples were rolled on blood agar plates and incubated for 16 h at 37 °C. Microbial growth was rated from 0 (no growth) to 5 (massive growth). No root canal obturation was performed but the access cavities were either restored with composite resin or left open for the remainder of the study period (histologic evaluation ongoing and not reported here).

## 3. Results

### 3.1. Determination of Disinfection Parameters

Extended rinsing with saline for 7.5 min alone led to a reduction in monospecies *E. faecalis* bacterial biofilm in the root canal, but it was not sufficient to completely clean the canal system. Electrochemical disinfection using BDD electrodes showed a time-dependent elimination of bacteria with 0.5 min application time being as effective as 7.5 min of saline rinsing. Already after applying BDD electrodes for 1 min, the mean of surviving bacteria was below 5% (Figure 3).

### 3.2. Multispecies Biofilm Formation

While biofilm formation on human root dentin slabs was reported already after 24 h for *E. faecalis* [39], data for multispecies biofilm was lacking. Therefore, we inoculated human teeth with a mixture of microorganisms isolated from root canal instruments [8,29] and incubated these for seven days at 37 °C. Biofilm formed was stained using EbbaBiolight680, a fluorescent dye interacting with the extracellular polysaccharide matrix of biofilm-forming microorganisms. Biofilm was analyzed at day 5 and 7 of incubation (Figure 4).

### 3.3. Elimination of Multispecies Biofilm

In a subsequent set of experiments, prepared teeth were incubated with bacterial suspension derived from root canal treatments and incubated at 37 °C for three days to induce multispecies biofilm formation. Following irrigation with saline as control or treatment with BDD electrodes, bacterial sampling using paper points was carried out as described above (see Section 2.2). Subsequently, the roots were split longitudinally, and the segments were pressed on blood agar plates, which were incubated overnight at 37 °C, as previously described [32].

Qualitative (impressions of split teeth on blood agar plates) and quantitative (sampling by paper points followed by incubation in BHI) experiments revealed similar results. Irrigation with saline did not reduce bacterial load significantly and resulted in approx. 50,000 CFU/mL. Application of CHX was more successful and treatment reduced the number to 50–500 CFU/mL. Compared to CHX, BDD treatment showed even better results. In this case, a complete elimination of bacteria was obtained in case of three replicates, one using 4.5 V and two using 6.5 V (Figure 5).

Identification of bacteria from selected colonies appearing after BDD treatment and incubation overnight at 37 °C revealed a higher resistance of Gram-positive bacteria towards electrochemical disinfection. Using mass spectrometry *Bacillus subtilis*, *Staphylococcus epidermidis* and *Staphylococcus xylosus* were identified as highly resistant species. 

### 3.4. Canine Tooth Model 

Nine teeth were conventionally treated, and eight teeth received additional BDD treatment. The mean values and standard deviations for rating bacterial growth on blood agar plates are given in Figure 6.

As expected, a massive reduction in bacterial growth was accomplished by endodontic treatment using hand instruments and conventional, non-activated irrigation with CHX. Additional application of BDD electrodes led to a further reduction in bacterial load. However, a complete sterilization of the canal system as observed with isolated teeth was not obtained. 

## 4. Discussion

Characteristic for biologic systems, an exponential elimination curve for bacterial biofilm was observed in all of the experiments conducted here. Conventional root canal treatment already seems to eliminate the majority of microbes present, while the additional use of electrochemical disinfection further reduces microbial load. For understanding the exact contributions of the different radicals produced from saline using BDD electrodes on disinfection, additional experiments are required.

Similarly to previous alternative approaches to root canal disinfection such as continuous apical negative ultrasonic irrigation [18,19], high-frequency electrical pulses [20], electric current and silver electrodes [21] as well as Er:YAG laser irradiation [22], BDD electrodes led to a decrease in bacterial count but could not predictably prevent bacterial regrowth [23] under clinical conditions.

NaOCl, the most widely used irrigant in endodontics [40] has not been used as a control in this series of experiments. While the antimicrobial effect of NaOCl is well documented, it has been shown that narrow, apical regions of root canals can hardly be reached by conventional irrigation [40]. While it may be seen as a limitation of the experiments not to include NaOCl as control, this was done in order to demonstrate the pure effect of BDD electrodes without “overshadowing” by a potent antimicrobial agent.

Electrophoresis is the only approach of applying electric current for root canal treatment the authors of this paper are aware of. While never reaching universal adaptation, good results have recently been reported for this modality [24,25]. It has to be emphasized that the electrodes described here have a different mode of action, i.e., the in situ generation of disinfective radicals from saline solution. As this approach is yet under development, no clinical experience exists so far.

Future studies require finding optimum performance parameters for BDD electrodes with respect to disinfection and potentially negative side effects to host tissue in the periapical region. The specimen derived from the animal experiment are currently subject to histologic analysis for evaluating the host response. Based on preliminary work using cell cultures, it can be expected that BDD electrodes do not cause more harm to living periapical tissues as compared to established irrigation methods. As such, the disinfection efficacy of BDD electrodes may not be limited by anatomical variation of the canal system [2] but instead may be time- and charge-dependent.

A major benefit of BDD electrodes could be that application of the electrodes is possible already at early stages of endodontic treatment when the instrumentation process is not yet advanced. The electrodes used here require an apical diameter of the root canal of 300 µm (ISO size #30) which is merely two sizes following establishment of a glide path. As such, BDD electrodes could ensure delivery of reactive oxygen species in the critical region at the apex, which can otherwise hardly be reached by conventional irrigation.

## 5. Conclusions

Based on the results of these preliminary studies presented here, the use of electrochemical disinfection with BDD electrodes following mechanical preparation of root canals may further reduce bacterial loads. First performance parameters, with respect to application time, irrigant volume and electric charge, have successfully been established for a prototype device. The disinfection system described here requires further optimization especially with respect to treatment time prior to clinical application.

## Figures and Tables

**Figure 1 microorganisms-10-00782-f001:**
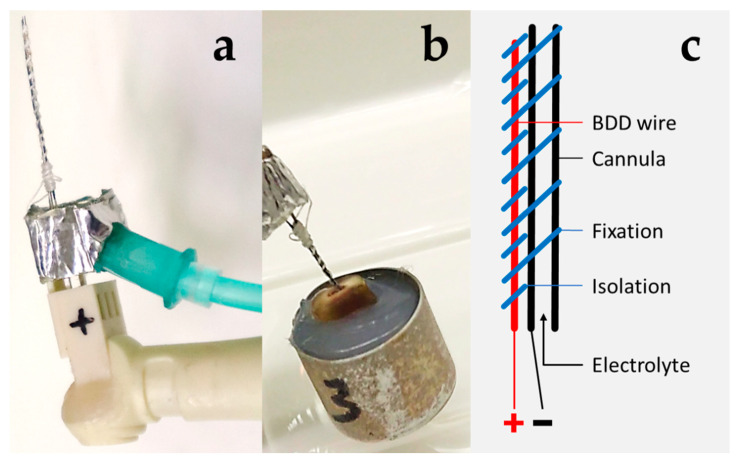
Determination of disinfection parameters using prototype 1. (**a**) Detailed view of prototype 1. A cannula for saline delivery was wired as cathode and combined with a BDD-coated niobium wire as anode. The handle seen on the left-hand side mimics a dental contra-angle and is used for connecting to a power and electrolyte source. (**b**) Application of prototype for disinfection of a root canal (extracted human tooth following crown removal at the cemento-enamel junction embedded in silicone). Note the saline flux from the root canal. (**c**) Scheme of electrode with anode (+) and cathode (−).

**Figure 2 microorganisms-10-00782-f002:**
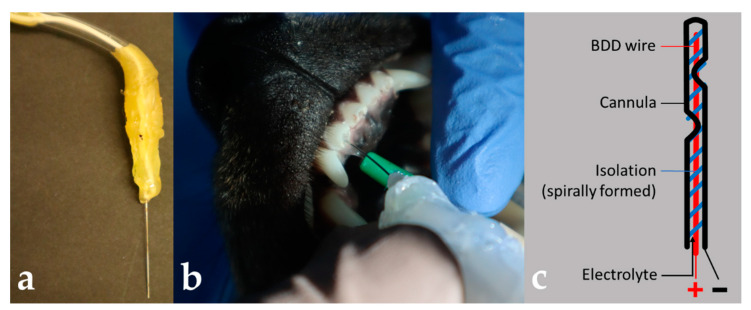
Canine tooth model for BDD electrode treatment using prototype 2. (**a**) Detailed view of prototype 2 electrode. A cannula wired as cathode was combined with an internally positioned BDD-coated niobium wire using insulating material. (**b**) Clinical situation showing the application of BDD electrode inside a root canal of a maxillary incisor. (**c**) Scheme of electrode with anode (+) and cathode (−).

**Figure 3 microorganisms-10-00782-f003:**
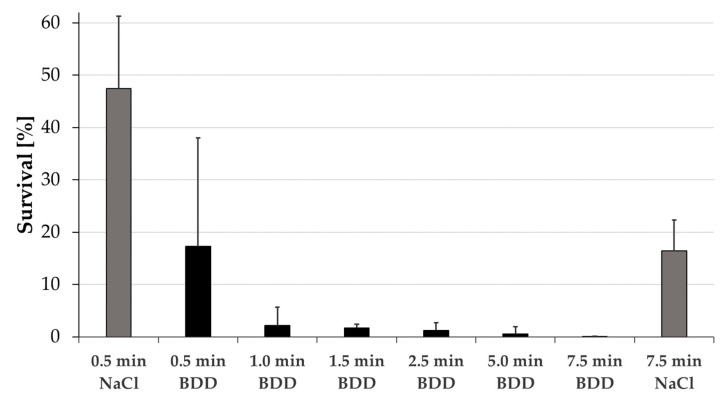
Disinfection of root canals. Percentage of *E. faecalis* surviving root canal irrigation with saline (grey) and electrochemical disinfection (black) for treatment times up to 7.5 min. CFU before treatment were set to 100%. All experiments were carried out in quadruplicate (independent biological replicates) and standard deviations are shown.

**Figure 4 microorganisms-10-00782-f004:**
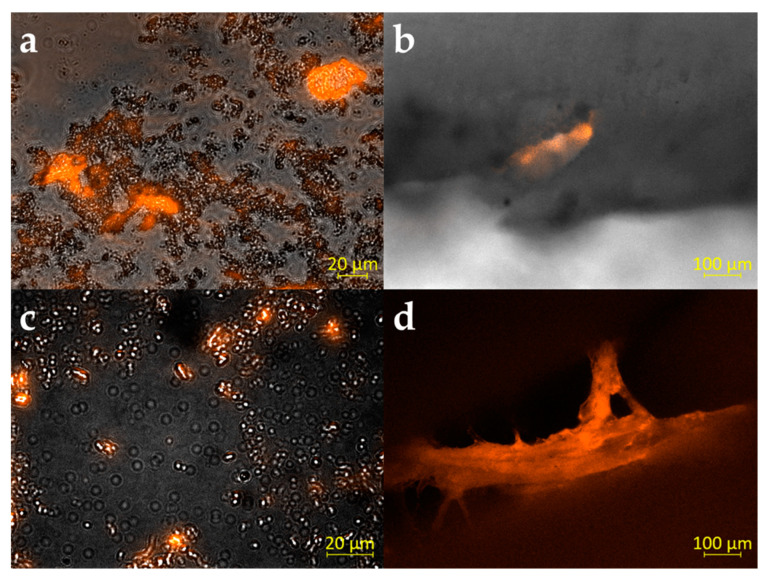
Microscopic imaging of biofilm formation. The extracellular polysaccharide matrix of biofilm-forming microorganisms was stained using EbbaBiolight680 after 5 (**a**,**b**) and 7 (**c**,**d**) days of incubation. Biofilm was either removed from the root canals before staining using toothpicks (**a**,**c**) or split teeth were analyzed (**b**,**d**). Panel b shows biofilm in a side channel and panel d shows biofilm attached to the root canal wall located at the bottom of the picture. (**a**–**c**) are overlays of brightfield and fluorescence, (**d**) show fluorescence in the red channel of the microscope only.

**Figure 5 microorganisms-10-00782-f005:**
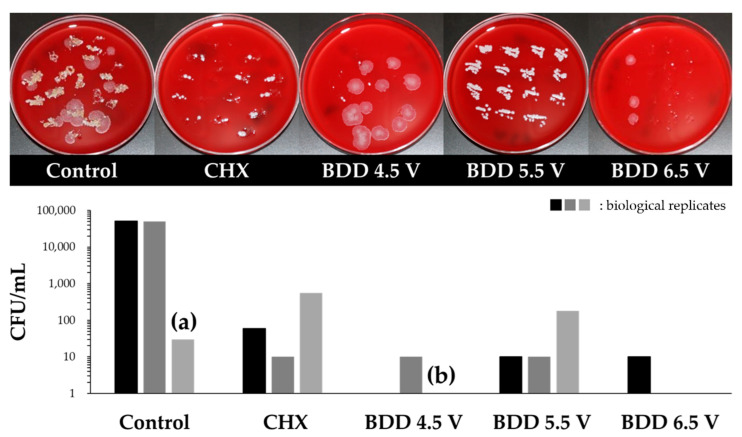
Elimination of multispecies biofilm. **Upper panel**: Bacterial growth on Columbia blood agar plates (typical examples are shown). **Lower panel**: Bacterial sampling using paper points and incubation in BHI. (**a**) technical problems during sampling, (**b**) condition only in duplicate.

**Figure 6 microorganisms-10-00782-f006:**
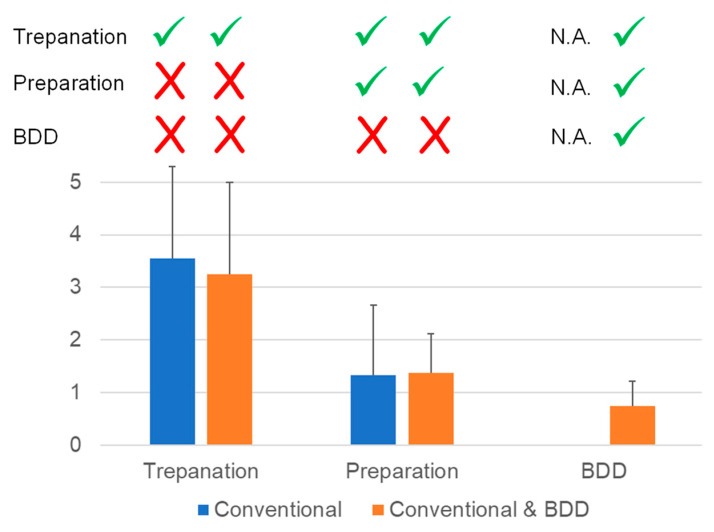
Ratings of bacterial growth following conventional endodontic treatment and endodontic treatment with additional BDD application. Electrochemical disinfection allowed for a further reduction in bacterial load. Meanings: ✓ Treatment step carried out; X Treatment step not carried out.

**Table 1 microorganisms-10-00782-t001:** Protocol for multispecies biofilm removal of infected root canals. Experimental groups (n = 3 roots per group) and final disinfection steps rendered following root canal instrumentation.

Group	Treatment
control	irrigation with saline (0.9% NaCl)
CHX	irrigation with chlorhexidine (0.2%)
BDD 4.5 V/0.5 mA	electrochemical disinfection and irrigation with saline
BDD 5.5 V/3.0 mA	electrochemical disinfection and irrigation with saline
BDD 6.5 V/10.0 mA	electrochemical disinfection and irrigation with saline

## Data Availability

Raw data will be provided by the authors on request.

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
