# Peer review of "Preclinical Testing of Boron-Doped Diamond Electrodes for Root Canal Disinfection—A Series of Preliminary Studies"

_microorganisms, 2022, doi:10.3390/microorganisms10040782_

Round 1
Reviewer 1 Report
Overall this is a very interesting paper. The following issues need to be addressed.
1. In the first paragraph of the introduction it would be worthwhile mentioning that the challenges include the organisms growing in the biofilm and also penetrating deep within tubules of the dentine.
2. Within figure 1 please include an electrical schematic of the overall approach and show clearly in a diagram how current flows between different parts of the apparatus inserted into the root canal.
3. In the text which immediately follows Figure 1 explain how the voltage and current parameters were chosen, and whether these were based on optimisation studies.
4. The analysis method for E. faecalis Uses overnight culture, and thus suffers from methodological issues, including carryover of antimicrobial agents from the electrolyte into the BHI (unlikely to be important because of the bulk dilation that occurs), and a culture time for the agar plates that is too small to overcome the effects of VBNC that occur with enterococci. The plates should have been scored after a longer culture period (e.g. 5 days) so that one could be completely sure that no slow-growing organisms were present. If the authors have not undertaken such prolonged cultures and they need to discuss the problem of VBNC as a limitation of their work. Many relevant points of the BNC are included in the following publication https://www.ncbi.nlm.nih.gov/pmc/articles/PMC4116801/ Note that because the same one-day culture period was used with the subsequent microbial assays that the issue of VBNC bacteria could also affect later parts of the study.
5. In line 120 of the methods please state which aerobic bacterial flora were included, and what culture medium was used. The incubation period of three days is likely to be insufficient for the development of the biofilm. Do the authors have evidence to present including confocal microscope or scanning electron microscope images that they have actually achieved a biofilm in such a short period of time?
6. In the methodology, include the details for procedures used for the identification of bacteria using mass spectrometry.
7. In relation to the discussion, the present study would have been much more informative if there was a direct comparison with the positive control of 1 or 4% NaOCl which is the standard antimicrobial irrigant used in endodontics around the world. The authors must comment on this point in the discussion. It is particularly relevant to the place of BDD as a potential treatment - is it an adjunct after NaOCl, or is it a complete replacement for NaOCl?
8. Given that the authors used a canine animal model, did they assess through histological analysis the extent of inflammatory changes in the periapical tissue resulting from chemical species generated by electrolysis of the saline? If this is a work in progress, then it should be mentioned.
9. The introduction or the discussion should mention the active species generated by electrolysis of saline, namely are the effects driven mostly by the hydroxyl ions generated or by the chlorine gas causing a physical dispersion of the biofilm?
10. It does not seem appropriate that the authors have stated Data Availability Statement: Not applicable. There should be CFU data available for different phases of the study.
Author Response
Reviewer #1
Overall this is a very interesting paper. The following issues need to be addressed.
Re.: Thank you very much for the positive response and the helpful comments.
- In the first paragraph of the introduction it would be worthwhile mentioning that the challenges include the organisms growing in the biofilm and also penetrating deep within tubules of the dentine.
Re.: This aspect has been added and we cited the following reference: Gänsbauer M, Burkovski A, Karl M, Grobecker-Karl T. Comparison of simplistic biofilm models for evaluating irrigating solutions. Quintessence Int. 2017;48(7):521-526. doi: 10.3290/j.qi.a38268. PMID: 28512652.
- Within figure 1 please include an electrical schematic of the overall approach and show clearly in a diagram how current flows between different parts of the apparatus inserted into the root canal.
Re.: A scheme of the respective electrode has been added to figures 1 and 2.
- In the text which immediately follows Figure 1 explain how the voltage and current parameters were chosen, and whether these were based on optimisation studies.
Re.: Determination of parameters was already described before. This information is given in line 119
- The analysis method for E. faecalis Uses overnight culture, and thus suffers from methodological issues, including carryover of antimicrobial agents from the electrolyte into the BHI (unlikely to be important because of the bulk dilation that occurs), and a culture time for the agar plates that is too small to overcome the effects of VBNC that occur with enterococci. The plates should have been scored after a longer culture period (e.g. 5 days) so that one could be completely sure that no slow-growing organisms were present. If the authors have not undertaken such prolonged cultures and they need to discuss the problem of VBNC as a limitation of their work. Many relevant points of the BNC are included in the following publication https://www.ncbi.nlm.nih.gov/pmc/articles/PMC4116801/ Note that because the same one-day culture period was used with the subsequent microbial assays that the issue of VBNC bacteria could also affect later parts of the study.
Re.: Thank you for this very valuable hint, which is especially interesting for further in vivo studies for us. However, Enterococcus faecalis was used in this study only to test if the new electrode design works comparable to former electrodes.
- In line 120 of the methods please state which aerobic bacterial flora were included, and what culture medium was used. The incubation period of three days is likely to be insufficient for the development of the biofilm. Do the authors have evidence to present including confocal microscope or scanning electron microscope images that they have actually achieved a biofilm in such a short period of time?
Re.: The medium used is given now (line 158). We apologize for a mistake in the manuscript. Bacteria were incubated for SEVEN days (corrected in line 159). To address the very valid point of the reviewer, we carried out additional experiments using a commercial fluorescence stain for biofilm detection.
- In the methodology, include the details for procedures used for the identification of bacteria using mass spectrometry.
Re.: Mass spectrometry was carried out by a commercial provider as indicated in the text (l. 173)
- In relation to the discussion, the present study would have been much more informative if there was a direct comparison with the positive control of 1 or 4% NaOCl which is the standard antimicrobial irrigant used in endodontics around the world. The authors must comment on this point in the discussion. It is particularly relevant to the place of BDD as a potential treatment - is it an adjunct after NaOCl, or is it a complete replacement for NaOCl?
Re.: A further paragraph has been added to the discussion section and we are citing the following reference: Haapasalo M, Shen Y, Wang Z, Gao Y. Irrigation in endodontics. Br Dent J. 2014 Mar;216(6):299-303. doi: 10.1038/sj.bdj.2014.204. PMID: 24651335.
- Given that the authors used a canine animal model, did they assess through histological analysis the extent of inflammatory changes in the periapical tissue resulting from chemical species generated by electrolysis of the saline? If this is a work in progress, then it should be mentioned.
Re.: Yes, we will evaluate periapical inflammation but this is ongoing, which has been added to the text.
- The introduction or the discussion should mention the active species generated by electrolysis of saline, namely are the effects driven mostly by the hydroxyl ions generated or by the chlorine gas causing a physical dispersion of the biofilm?
Re.: Based on the experiments carried out, we cannot distinguish the contributions of the different radicals produced from saline causing this effect.
- It does not seem appropriate that the authors have stated Data Availability Statement: Not applicable. There should be CFU data available for different phases of the study.
Re.: We changed the data availability statement accordingly (l. 340).
Reviewer 2 Report
General comment: The authors have covered and described an interesting topic and in general it is a beautiful article! But in some places, it is not clear what they wanted to turn to due to the chosen English, that the latter is confused. Please find critical comments as follows:
Critical comments and suggestions: Since this is a short paper, the referee suggests improving the title by changing to that form to attract the reader more when deepening into the text by better demonstrating one's work. It may be that in the title can be added "preliminary study... or pilot study...", to be in line with the experiments carried out. Looking at the abstract, in lines 20-22, the authors begin their experiment with extracted human teeth and then move on to the canine model. The aim of this work is unclear and the authors are confused in expressing themselves. At this point, the abstract must be corrected after major revision of this paper!
- Introduction: This section needs to be reviewed following a chronological path. Being an “article” and not a “short communication”, the authors must explore more in the other methods using electrodes in endodontic treatments, with the aim of inhibition of microbial biofilms and intracanal disinfection. Here below are two articles to be including in the introduction section for comparison: [-Antimicrobial and antibiofilm efficacy of a copper/calcium hydroxide-based endodontic paste against Staphylococcus aureus, Pseudomonas aeruginosa and Candida albicans. Dent Mater J. 2019 Jul 31;38(4):591-603. doi: 10.4012/dmj.2018-252.; -Copper-Calcium Hydroxide and Permanent Electrophoretic Current for Treatment of Apical Periodontitis. Materials (Basel). 2021 Feb 2;14(3):678. doi: 10.3390/ma14030678. PMID: 33540551].
- In line 33, the beginning of the sentence doesn’t sound scientifically good in English. The same is noticed in line 54 for “wide body”…
- Lines 42-46 aren’t clear in what they represent.
- The aim must be clear and the lines 63-67 are a mess, not divided in respective points that are followed below in the respective sections and subsections.
- Materials and Methods
- In subsection 2.1, if it is possible to make a scheme or flowchart it would be nice to better orient the reader.
- Lines 85 and 113: The referee suggests to choose another word for “decapitated”, may be “cut”.
- Line 88: Why did the authors use chlorhexidine rather than sodium hypochlorite as an irrigant?
- Figures 1 & 2: Please write smaller "a" and "b" within their respective photos and replace them with clear ones, as they are both blurry images.
- Line 112: What is the difference in "Wild Type Biofilm" that the authors used here as a notion? What does it concretely represent… the authors must specify by describing it better or by choosing another term.
- Line 122: Why did you choose three days instead of seven for bacterial inoculation? Did you use incubation at night under shaking? How did you prove after this step that bacterial inoculation was sufficient after the three days?
- Line 125: Please better describe this kind of root placement procedure.
- Line 132: Instead of “chapter”, you can use “subsection 2.2.”.
- Line 135: Put the long name of MALDI-TOF first and then its acronym in parentheses.
- Results
- Lines 182 & 189: Express in italics the bacterial name.
- Figures 3, 4 & 5 must be replaced with clearer ones of better quality.
- Lines 214-215: Express in italics the bacterial names.
Pay attention to the English language used here, to correct it adequately with the scientific one.
- Discussion: This section is too short to be a discussion of an article, it looks more like a discussion of a short communication, it should include more information while discussing other papers.
- Conclusions: Furthermore, this paragraph needs to be revised and mention the correct conclusions of this pilot study.
Author Response
Reviewer #2
General comment: The authors have covered and described an interesting topic and in general it is a beautiful article! But in some places, it is not clear what they wanted to turn to due to the chosen English, that the latter is confused. Please find critical comments as follows:
Re.: Thanks much for this comment! We have properly revised the paper ensuring to increase clarity.
Critical comments and suggestions: Since this is a short paper, the referee suggests improving the title by changing to that form to attract the reader more when deepening into the text by better demonstrating one's work. It may be that in the title can be added "preliminary study... or pilot study...", to be in line with the experiments carried out.
Re.: We have added “-a series of preliminary studies” as subtitle hoping that this is acceptable
Looking at the abstract, in lines 20-22, the authors begin their experiment with extracted human teeth and then move on to the canine model. The aim of this work is unclear and the authors are confused in expressing themselves. At this point, the abstract must be corrected after major revision of this paper!
Re.: We are aware that a series of interdependent experiments is difficult to explain in a short abstract. The respective section has now been modified.
Introduction: This section needs to be reviewed following a chronological path. Being an “article” and not a “short communication”, the authors must explore more in the other methods using electrodes in endodontic treatments, with the aim of inhibition of microbial biofilms and intracanal disinfection. Here below are two articles to be including in the introduction section for comparison: [-Antimicrobial and antibiofilm efficacy of a copper/calcium hydroxide-based endodontic paste against Staphylococcus aureus, Pseudomonas aeruginosa and Candida albicans. Dent Mater J. 2019 Jul 31;38(4):591-603. doi: 10.4012/dmj.2018-252.; -Copper-Calcium Hydroxide and Permanent Electrophoretic Current for Treatment of Apical Periodontitis. Materials (Basel). 2021 Feb 2;14(3):678. doi: 10.3390/ma14030678. PMID: 33540551].
Re.: Thank you very much for pointing us to these highly valuable reports, which we have included in the reference list. Kindly note that our approach differs considerably from electrophoresis!
- In line 33, the beginning of the sentence doesn’t sound scientifically good in English. The same is noticed in line 54 for “wide body”…
Re.: Both sentences have been modified
- Lines 42-46 aren’t clear in what they represent.
Re.: Sentence has been reworded
- The aim must be clear and the lines 63-67 are a mess, not divided in respective points that are followed below in the respective sections and subsections.
Re.: This section has been reworded for clarity
Materials and Methods
- In subsection 2.1, if it is possible to make a scheme or flowchart it would be nice to better orient the reader.
Re.: A scheme of the electrode is given in Figure 1 c now. We did not add a figure of the coating process, since in our feeling this is far beyond the focus of microorganisms.
- Lines 85 and 113: The referee suggests to choose another word for “decapitated”, may be “cut”.
Re.: Replaced “decapitated” with “cut” throughout the manuscript
- Line 88: Why did the authors use chlorhexidine rather than sodium hypochlorite as an irrigant?
Re.: This point has also been raised by reviewer #1. We have added to the discussion that this was done in order not to overshadow the effect of BDD mediated disinfection
- Figures 1 & 2: Please write smaller "a" and "b" within their respective photos and replace them with clear ones, as they are both blurry images.
Re.: Done.
- Line 112: What is the difference in "Wild Type Biofilm" that the authors used here as a notion? What does it concretely represent… the authors must specify by describing it better or by choosing another term.
Re.: Wild Type Biofilm was replaced by multispecies biofilm. Its isolation from root canal treatment instruments was described before and is cited in 2.4 (line 155)
- Line 122: Why did you choose three days instead of seven for bacterial inoculation? Did you use incubation at night under shaking? How did you prove after this step that bacterial inoculation was sufficient after the three days?
Re.: We apologize for a mistake in the manuscript. Bacteria were incubated for SEVEN days (corrected in line 159). To address the very valid point both reviewers, we carried out additional experiments using a commercial fluorescence stain for biofilm detection.
- In the methodology, include the details for procedures used for the identification of bacteria using mass spectrometry.
Re: Mass spectrometry was carried out by a commercial provider as indicated in the text (l. 173)
- Line 125: Please better describe this kind of root placement procedure.
Re.: Done (line 162); This was a simple handling aspect and the setup was comparable to Fig. 1b.
- Line 132: Instead of “chapter”, you can use “subsection 2.2.”.
Re.: Changed accordingly.
- Line 135: Put the long name of MALDI-TOF first and then its acronym in parentheses.
Re.: Done (line 172)
Results
- Lines 182 & 189: Express in italics the bacterial name.
Re.: Done (line 221, 228)
- Figures 3, 4 & 5 must be replaced with clearer ones of better quality.
Re.: We apologize for the quality of the line graphics but even after re-checking the capabilities at our universities we are unfortunately not able to provide better quality. Thanks for your understanding.
- Lines 214-215: Express in italics the bacterial names.
Re.: Corrected (line 270, 271)
Pay attention to the English language used here, to correct it adequately with the scientific one.
Re.: The whole manuscript has been rechecked by a native English speaker proficient in scientific writing. If there are remaining concerns, the reviewer is kindly asked to point these out.
Discussion: This section is too short to be a discussion of an article, it looks more like a discussion of a short communication, it should include more information while discussing other papers.
Re.: We have added a further paragraph on electrophoresis as a reference system also applying electric current during root canal treatment.
Conclusions: Furthermore, this paragraph needs to be revised and mention the correct conclusions of this pilot study.
Re.: We have reworded and elaborated on the Conclusions
Round 2
Reviewer 2 Report
The authors improved their manuscript respecting the comments made! The referee has only a few things to comment: the first is on line 74 where it is necessary to correct "this series" in "these series"; the second in line 75 where "in vitro" must be written in italics; other, please decrease the size of the letters inside the figures 1,2,4, which are huge; the last, the authors have to explain better in line 162 the procedure used for the "sterile plasticine ...". The latter creates confusion as to how it was used being a soft modeling material ...? It would be nice to insert a photo or making a scheme of how it was used.
Author Response
The authors improved their manuscript respecting the comments made!
The referee has only a few things to comment: the first is on line 74 where it is necessary to correct "this series" in "these series";
Re.: Changed (l. 73/74).
the second in line 75 where "in vitro" must be written in italics;
Re.: Done (l. 74/75).
other, please decrease the size of the letters inside the figures 1,2,4, which are huge;
Re.: Changed as recommended by the reviewer.
the last, the authors have to explain better in line 162 the procedure used for the "sterile plasticine ...". The latter creates confusion as to how it was used being a soft modeling material ...? It would be nice to insert a photo or making a scheme of how it was used.
Re.: A new reference including the requested information was added (l. 162, ref [38]).
All changes made are marked in yellow.